# Sol–Gel-Processed Organic–Inorganic Hybrid for Flexible Conductive Substrates Based on Gravure-Printed Silver Nanowires and Graphene

**DOI:** 10.3390/polym11010158

**Published:** 2019-01-17

**Authors:** Xinlin Li, Nahae Kim, Seongwook Youn, Tae Kyu An, Juyoung Kim, Sooman Lim, Se Hyun Kim

**Affiliations:** 1Department of Mechanical Engineering Science, Yeungnam University, Gyeongsan, Gyeongbuk 38541, Korea; xinlin0618@gmail.com; 2Department of Advanced Materials Engineering, Kangwon National University, Samcheok 25931, Korea; kjr2856@naver.com; 3Department of Software, Korea National University of Transportation, 50 Daehak-Ro, Chungju 27469, Korea; youn@ut.ac.kr; 4Department of Polymer Science and Engineering & IT Convergence, Korea National University of Transportation, 50 Daehak-Ro, Chungju 27469, Korea; 5Graduate School of Flexible and Printable Electronics, Chonbuk National University, Jeonju 54896, Korea; 6School of Chemical Engineering, Yeungnam University, Gyeongsan, Gyeongbuk 38541, Korea

**Keywords:** conductive substrate, gravure printing, flexible electronics, and nanomaterials

## Abstract

In this study, an organic–inorganic (O–I) nanohybrid obtained by incorporating an alkoxysilane-functionalized amphiphilic polymer precursor into a SiO_2_–TiO_2_ hybrid network was successfully utilized as a buffer layer to fabricate a flexible, transparent, and stable conductive substrate for solution-processed silver nanowires (AgNWs) and graphene under ambient conditions. The resulting O–I nanohybrid sol (denoted as AGPTi) provided a transmittance of the spin-coated AgNWs on an AGPTi-coated glass of 99.4% and high adhesion strength after a 3M tape test, with no visible changes in the AgNWs. In addition, AGPTi acted as a highly functional buffer layer, absorbing the applied pressure between the conductive materials, AgNWs and graphene, and rigid substrate, leading to a significant reduction in sheet resistance. Furthermore, gravure-printed AgNWs and graphene on the AGPTi-based flexible substrate had uniform line widths of 490 ± 15 and 470 ± 12 µm, with 1000-cycle bending durabilities, respectively.

## 1. Introduction

Recently, flexible electronic devices, such as displays, energy harvesting systems, memories, sensors, and many more [1,2,3,4,5,6,7], have attracted significant interest. Intensive studies have led to the development of prototype flexible devices, enabled by the fabrication of novel materials, development of processing techniques, and simulations on the device physics. However, several scientific and technical challenges should be overcome to commercialize the devices [8,9]. In particular, future flexible electronics will require solution-based printing processes instead of vacuum and photolithographic processes, which have been widely used in the fabrication of the current electronic devices [10,11,12,13,14,15]. The printing processes enable direct patterning of inks on various substrates (e.g., glass, polymer, metal foil, and paper) without auxiliary deposition and removal of a photoresist, which provides valuable advantages including simplicity, low cost, and large-area solution processability. Therefore, significant progress is required in the development of printable materials for flexible electronic devices [16,17,18,19,20].

As electronic devices are fabricated on substrates, flexible conductive substrates are the basis for the successful fabrication of flexible electronics. Among the flexible conductive substrates, indium tin oxide (ITO)-coated polymers (such as polyethylene terephthalate (PET), polyethylene naphthalate (PEN), and poly(ethylene succinate) (PES)) have been widely used owing to the high transparency and conductivity of the ITO layer [21,22,23,24,25]. However, there are disadvantages, as ITO is rare, brittle, and requires vacuum processing, which can hinder its applicability to the rollable electronics. Therefore, alternative conductive materials, such as carbon nanomaterials, metal nanoparticles and nanowires, and organic conducting polymers, are extensively investigated through solution processes [26,27,28,29,30].

Prior to the deposition of conductive materials on a polymer substrate, the introduction of a buffer layer is essential to improve surface characteristics of the polymer substrate to fabricate a smooth surface, and to fill in pinholes and to enhance the adhesion between the conductive materials and substrate [31,32,33]. In addition, the buffer layer should improve the mechanical, thermal, and chemical stabilities of the underlying substrate [34,35,36]. Various polymers can be easily formed as buffer layers through roll-to-roll and printing processes. However, their resistance to mechanical impacts (e.g., abrasion) and excessive thermal and chemical treatments still needs to be improved. An inorganic coating fabricated by gas-phase vacuum deposition (chemical vapor deposition (CVD), physical vapor deposition (PVD)) provides excellent abrasion strength as well as thermal and chemical resistance. However, several factors, such as the high cost, weak adhesion, and brittleness, hinder its application in flexible electronics [37].

O–I nanohybrid materials were utilized as they exhibit desirable characteristics of both inorganic (thermal stability, mechanical integrity, etc.) and organic (flexibility, simple processing, specific chemical functionality, etc.) materials [38,39,40,41]. In particular, the sol–gel-derived O–I nanohybrids enabled simple processing. Depending on the choice of precursors and their combinations, it is possible to obtain nanohybrid films with variable physicochemical properties. Therefore, the sol–gel deposition of thin films has been applied to various fields owing to the advantages such as abrasion resistance, anticorrosion, antireflection, and self-cleaning [42]. These advantages are attributed to the stable covalent bonds formed at the interface between the O–I nanohybrid and substrate with surface hydroxyl groups. The stable covalent bonds originate from the sol–gel reaction route. Common sol–gel reaction routes during the sol–gel process are:

Si–(OR)*_n_* + H_2_O ↔ Si–OH + ROH: hydrolysis,
(1)

Si–OH + Si–OH ↔ Si–O–Si + H_2_O: water condensation,
(2)

Si–OH + Si–OR ↔ Si–O–Si + ROH: alcohol condensation.
(3)


Substitution of the hydroxyl groups (–OH) for alkoxide groups (–OR) of the silane compound occurs during the hydrolysis, owing to the interaction of alkoxide groups with H_2_O; subsequently, water condensation or/and alcohol condensation occur, producing Si–O–Si covalent bonds, water, and alcohol, respectively. The condensations occur not only in the O–I nanohybrid bulk but also at the interface with the substrate. Although alkoxysilane-based O–I nanohybrids are advantageous in several applications, some issues need to be overcome to enable commercialization. For example, O–I hybrid sols are prepared using only alkoxysilane compounds, often suffering from aggregation, which implies that the gelation reaction unexpectedly occurs during the film processing. In order to address the gelation issue, amphiphilic molecules have been used as a dispersion agent to obtain stable sols. However, amphiphilic materials, such as the commonly used surfactants, do not participate in the reaction. Therefore, they should be removed after the preparation to avoid deterioration in the mechanical properties of the O–I hybrid materials [43]. In this regard, it is required to synthesize amphiphilic materials that can be chemically bonded to alkoxysilanes, yielding stable and mechanically strong O–I hybrid materials. The alkoxysilane-functionalized amphiphilic polymer (AFAP) polymer used in this study consists of polyethylene oxide-based hydrophilic and polypropylene oxide-based hydrophobic segments with reactive alkoxysilane groups at both ends. It could be expected that the amphiphilic polymer backbones of AFAP colloidally stabilize O–I nanohybrid sols formed during the hydrolytic condensation polymerization without a dispersion agent. Moreover, their alkoxysilane groups can chemically react with organosilanes, thereby leading to Si–O–Si networks, which can improve the miscibility between the organic and inorganic components in the resulting O–I nanohybrid materials. Simultaneously, the amphiphilic polymer backbone of the alkoxysilane precursor facilitates the formation of a flexible homogeneous film after the curing process.

In this study, in order to fabricate a flexible conductive substrate based on solution-processed silver nanowires (AgNWs) and graphene, we demonstrate an organic–inorganic (O–I) nanohybrid by incorporating an alkoxysilane-functionalized amphiphilic polymer (AFAP) precursor into a SiO_2_–TiO_2_ hybrid network. The resulting O–I nanohybrid sol can be deposited to a uniform film on substrates, such as glass and PET, by spin- and bar-coating methods. Subsequently, annealing (105 °C for 30 min) hardens the film to achieve chemical and mechanical strength and transparency. AgNWs and graphene conductive materials are formed on the O–I nanohybrid-treated substrates by spray-coating and gravure printing, which enable fabrication of the micropatterns of the materials. The O–I nanohybrid-coated flexible conductive substrates exhibit high conductivities and good adhesion properties in both cases of AgNWs and graphene, with excellent bending stabilities at a radius of curvature of 5 mm.

## 2. Experimental Methods

### 2.1. Synthesis of the O–I Nanohybrid

In the synthesis of AFAP precursors [38], isophorone diisocyanate (IPDI, Aldrich Chemical Co., St. Louis, MS, USA), glycerol (Gc, *M*_w_: 92.09 g/mol, Aldrich Chemical Co., St. Louis, MS, USA), (3-aminopropyl)triethoxysilane (APTES, Aldrich Chemical Co.), and poly(ethylene glycol) (PEG, *M*_w_ = 300 g/mol, Aldrich Chemical Co., St. Louis, MS, USA) were used as received, without further purification. Dibutyltin dilaurate (DBTDL, Aldrich Chemical Co., St. Louis, MS, USA) was used as a catalyst in the precursor synthesis. (3-Glycidyloxypropyl)trimethoxysilane (GPTMS, 9%, Aldrich Chemical Co., St. Louis, MS, USA), propyltrimethoxysilane (PTMS, 97%, Aldrich Chemical Co., St. Louis, MS, USA), titanium(IV) isopropoxide (TTiP, 97%, Aldrich Chemical Co., St. Louis, MS, USA), ethanol (Aldrich Chemical Co., St. Louis, MS, USA), and hydrochloric acid (HCl, ACS reagent, 37%, Aldrich Chemical Co., St. Louis, MS, USA) were used, as received, for the preparation of O–I hybrid sols. Gc and PEG were dried under a reduced pressure for 24 h at 80 °C before use. The specific synthesis of the O–I nanohybrid colloidal solution is presented in a previous report [39]. In order to prepare the O–I nanohybrid sols, the AFAP precursors were mixed with GPTMS, PTMS, TTiP, ethanol, and water (0.1 M HCl (aq)), at a mass ratio of 1:1:1:4:0.4, and stirred at 60 °C for 48 h to induce the hydrolytic hydrolysis–condensation reaction. The resultant O–I hybrid sol is referred to as AGPTi, whereas the O–I hybrid sol without AFAP is denoted as GPTi. During the reaction, no macrophase separation and precipitation occurred in the solution without a chelating agent or stabilizer. 

### 2.2. Fabrication of Conductive Films

AgNWs and water-dispersed graphene were used as conductive materials in this study. The AgNWs were synthesized using ethylene glycol (EG) in the presence of ZnCl_2_, Fe(NO_3_)_3_∙H_2_O, and poly(*N*-vinylpyrrolidone), as reported in [40]. The resulting AgNWs had diameters of 40–80 nm and lengths of 30–80 μm. The prepared AgNWs (0.06 wt %) were dispersed in isopropyl alcohol (IPA) for coating onto glass and O–I nanohybrid films. A water-dispersible graphene paste ink was purchased from MExplorer (Ansan, Korea), in which graphene flakes are homogenously dispersed in deionized (DI) water without any organic binders. Bare glass (3.5 × 3.5 cm^2^) substrates were cleaned in boiled acetone and subsequently sonicated several times in acetone/isopropyl alcohol/distilled water. The cleaned bare glass was exposed to a UV ozone cleaner for 5 min for further hydrophilization. In order to fabricate transparent conductive thin films, dispersions of 50 mL of AgNWs (AgNWs/bare glass) and 100 mL of graphene mixed with IPA (volume ratio: 1:1) (graphene/bare glass) were spray-coated on bare glasses, used as references. For functionalization of the conductive thin film, both AGPTi and GPTi O–I nanohybrid sols mixed with propylene glycol monomethyl ether acetate (PGMEA, volume ratio: 1:1) were spun on a clean bare glass at 9000 rpm for 3 min at room temperature, followed by annealing at 105 °C for 30 min, as shown in Figure 1. On the O–I nanohybrid films on the bare glasses, the AgNW (AgNWs/O–I nanohybrid/glass) and graphene (graphene/O–I nanohybrid/glass) dispersions were spray-coated under the same conditions used for the references (Figure 1a). The conductive films were then mechanically pressed (P) at 10 MPa for 10 min to increase their conductivities. Finally, electrodes with low surface roughnesses, high transparencies, and low resistances were fabricated, with AgNWs (P-AgNWs/O–I nanohybrid/glass) and graphene (P-graphene/O–I nanohybrid/glass), as shown in Figure 1b. In order to apply these materials to gravure printing (G), the purchased AgNWs and graphene ink mixed with IPA (60 wt %) were employed. Optimization of printing parameters, such as ink viscosity, surface energy, and printing speed, enabled printing of conductive lines of AgNWs and graphene on both AGPTi- and GPTi O–I nanohybrid-coated PET substrates (G-AgNWs/O–I nanohybrids/PET). In this process, a copper gravure pattern manufactured by photolithography and chemical etching was utilized as a gravure pattern in trenches.

### 2.3. Characterization of the Conductive Films

In order to characterize their optical and electrical properties, the transmittances of the AgNW- and graphene-based conductive films were measured using a spectrophotometer (GENESYS 10S UV–vis, Thermo Fisher Scientific^TM^ (Seoul, Korea), showing spectral bandwidth of 1.8 nm and wavelength accuracy of ±1.0 nm), in which samples were measured in a range of wavelengths from 190 to 1100 nm. The sheet resistances (Rs) were measured using the four-probe technique (Loresta EP MCP-T360, Mitsubishi chemical, Tokyo, Japan, determination range: 10^−2^–10^6^ Ω). Optical microscopy (OM, Nikon ECLIPSE LV100ND, Nikon, Tokyo, Japan, Max. sample size: 150 × 150 mm^2^). Atomic force microscopy (AFM, Veeco DI Dimension 3100 with Nanoscope V, Al-coated Si tips which were purchased from Nanosensors^TM^, Neuchâtel, Switzerland) was used to measure surface properties. Tapping-mode was used to take AFM images with Al-coated Si tips which were purchased from NanosensorsTM (Neuchâtel, Switzerland, thickness: 4.0 ± 1 μm; length: 125 ± 10 μm; width: 30 ± 7.5 μm; resonance frequency: 204–497 kHz; force constant: 10–130 N/m; tip height: 10–15 μm). Scanning electron microscopy (SEM, Hitachi S4800, Hitachi, Tokyo, Japan, resolution: 1.0 nm at 15 kV, 1.4 nm at 1 kV, deacceleration mode) were used to measure the surface roughness and topography. In order to determine the mechanical properties, a taping test with a 3M tape was applied to the coated films, which were pressed evenly and simultaneously detached.

## 3. Results and Discussion

In order to investigate the effect of AFAP on the physical and chemical properties of the SiO_2_–TiO_2_-based O–I nanohybrid for the conductive substrate, AGPTi (with AFAP) and GPTi (without AFAP) O–I nanohybrid sols were coated onto glass and PET substrates through spin-casting and gravure printing. First, we characterized the transmittances of the glass substrates coated with the AGPTi and GPTi O–I nanohybrids, as a high transparency is very desirable in optoelectronic devices. Figure 2 shows the transmittances of the spin-coated AGPTi/glass (99.4% at 550 nm) and GPTi/glass (99.2% at 550 nm), which are comparable to that of the bare glass. In addition, both AGPTi/glass and GPTi/glass did not become hazy or degrade upon light irradiation. In general, nanostructured thin films can be efficiently deposited by spray-coating, providing a cheap large-area deposition of AgNWs and graphene [44,45]. Therefore, the fabrications of conductive thin films of AgNWs and graphene were performed on the samples by simple spray-coating processes under ambient conditions, as shown in Figure 1.

In order to investigate their adhesive strengths, 3M taping tests were performed for the thin films on the bare glass and AGPTi- and GPTi-coated glass substrates. The results showed low adhesions of AgNWs/glass (Figure 3a,a-1) and graphene/glass (Figure 3b,b-1). This tendency is generally observed for noble metals with weak adhesions on glass [46,47]. By contrast, the substrates functionalized by the deposited AGPTi and GPTi O–I nanohybrid layers had higher adhesion strengths; no visible changes in the AgNWs (Figure 3c,c-1 for AGPTi and Figure 3e,e-1 for GPTi) and graphene (Figure 3d,d-1 for AGPTi and Figure 3f,f-1 for GPTi) were observed after the taping test. The AGPTi and GPTi O–I nanohybrids had an important role as buffer matrices against the applied pressure. The AgNWs and graphene penetrated these polymer layers, leading to an enhancement in adhesion and, thus, to maintenance of sheet resistance after the taping test. Moreover, the pressed AgNWs and graphene on AGPTi and GPTi led to reductions in their sheet resistances, from 5 to 1.5 Ω/sq and from 700 to 600 Ω/sq, respectively (Figure 4). This is mainly attributed to the enhanced connection of the wire-to-wire junctions of the AgNWs during the pressing process [48]. In the case of graphene, the application of pressure enlarges the contact area between graphene particles, leading to a significant reduction in the additional resistance attributed to individual particles [49]. These results are demonstrated by optical microscopy (OM) and scanning electron microscopy (SEM) (insets) images of the AgNWs and graphene before (Figure 5a,c) and after (Figure 5b,d) the pressing process. A similar tendency was observed for the GPTi O–I nanohybrid-coated film, which is not shown in this figure. 

An AFM analysis (Figure 6) demonstrated that the pressing processes on the AGPTi and GPTi O–I nanohybrid films contributed to the decreases in the surface roughnesses of the AgNWs and graphene. In general, AgNWs exhibit a high surface roughness due to their intrinsic wire structure. The root-mean-square (*R*_q_) roughness of AgNWs/glass was 15 nm, corresponding to the diameters of the wires (60–100 nm). *R*_q_ significantly decreased to 2.5 nm after the pressing processes on the AGPTi (Figure 6a,a-1) and GPTi (Figure 6b,b-1) O–I nanohybrids, without sticking-out and loss of transparency. In a previous study, embedment of AgNWs on a predeposited film (e.g., oly(3,4-ethylenedioxythiophene):poly(styrenesulfonate) (PEDOT:PSS)) led to a decrease in surface roughness [50]. However, this affected the transparency owing to the trade-off relationship considering the original color of the conductive polymer [50]. In this study, this issue was overcome using the highly transparent films of the AGPTi and GPTi O–I nanohybrids, enabling the decreases in the surface roughnesses without loss of transparency. A similar trend was observed for the case of graphene. The roughness of graphene on the AGPTi (Figure 6c,c-1) and GPTi (Figure 6d,d-1) O–I nanohybrids (on glass) decreased from 13 to 1.5 nm after the pressing process. Although a study has been performed to reduce the surface roughness of graphene oxide (rGO) using a polyethersulfone film [51], it was challenging to fabricate a graphene film with high surface smoothness on a rigid substrate, owing to the brittleness of the rigid glass.

The AGPTi and GPTi O–I nanohybrid layers were further investigated to determine their suitability for roll-to-roll-based printed electronics using gravure printing. Gravure printing is a promising roll-to-roll-based process providing a high-speed deposition of functional materials in a large area with low cost, thus enabling very high productivity [52]. Fundamentally, this method involves three steps: filling of the pattern with ink, removal of excess ink from the surface, and transfer of the ink to the substrate. Therefore, an investigation of the relationship between the ink and substrate is necessary to improve the pattern fidelity. Using the optimized AgNWs and graphene ink for the gravure printing process, conductive lines of AgNWs (Figure 7a for AGPTi and Figure 7b for GPTi) and graphene (Figure 7c for AGPTi and Figure 7d for GPTi) were fabricated at a printing speed of 3 cm/s. The continuous line morphologies demonstrate the proper wettability and printability of the AGPTi and GPTi O–I nanohybrid layers to the desired materials for patterning. Trenches of 500 µm yielded lines of gravure-printed AgNWs/AGPTi/PET with widths of 490 ± 15 µm and gravure-printed graphene/AGPTi/PET with widths of 470 ± 12 µm. However, narrower lines were obtained on the GPTi O–I nanohybrid layer, with widths of 481 ± 15 µm (gravure-printed AgNWs/GPTi/PET) and 424 ± 15 µm (gravure-printed graphene/GPTi/PET), as shown in Figure 8a. This can be explained in terms of surface energy affecting the spreading of the water-based solution. As the contact angle of DI water on the GPTi O–I nanohybrid was 91.3°, which is 12° higher than that for the AGPTi nanohybrid, a lower surface energy induced by the lower polar energy was expected, leading to the reduced line width. In order to reveal the electrical properties of the printed AgNW and graphene lines, conductivity measurements were performed. The results of line resistance measurements are shown in Figure 8b. All of the printed lines (500–325 µm) were electrically continuous over the entire line lengths of 2 cm with a standard deviation lower than 10%, indicating their high continuity and uniformity. In addition, the tendency of decrease in line resistance as a function of the line width is attributed to the increase in electrical area for electron transport. Decreases in the line resistances, from 32 to 1 kΩ/mm for the gravure-printed AgNWs/GPTi/PET and from 25 to 0.4 kΩ/mm for the gravure-printed graphene/AGPTi/PET, were observed.

The high flexibilities of the films with the AGPTi and GPTi O–I nanohybrids were demonstrated by characterizing mechanical properties upon measurements of the line resistances over a large number of bending cycles. Figure 9a shows the normalized line resistances as a function of the bending cycle; no measurable loss of line resistance over the entire number-of-cycles range was observed for the AGPTi O–I nanohybrid (Figure 9b,c) at a radius of curvature of 5 mm. By contrast, a significant decrease in line resistance was observed for the GPTi O–I nanohybrid during the 1000 bending cycles. In particular, the gravure-printed AgNWs/GPTi/PET (Figure 9d) had a normalized resistance over 10 at 100 bending cycles, which was ten times higher than that of the gravure-printed graphene/GPTi/PET (Figure 9e). This is attributed to the easier breaking of AgNWs under the bending pressure. The origin of the higher endurance of the AGPTi O–I nanohybrid against the bending pressure (Figure 9d,e), than that of the GPTi nanohybrid, is attributed to the chemical structure of AFAP in the AGPTi O–I nanohybrid. As mentioned above, AFAP consists of two types of polymer chains: polyethylene oxide-based hydrophilic and polypropylene oxide-based hydrophobic. The amphiphilic AFAP could colloidally stabilize the O–I nanohybrid sols during the hydrolytic condensation polymerization. In the thin-film state, the polymer segment of AFAP can provide the flexibility between O–I nanohybrid particles forming the film, which contributed to the endurance of the AGPTi O–I nanohybrid against the bending pressure. Therefore, AGPTi has desirable mechanical properties enabling the fabrication of highly functional flexible devices [53].

## 4. Conclusions

AGPTi, an O–I nanohybrid obtained by incorporating the AFAP precursor into the SiO_2_–TiO_2_ hybrid network, was successfully utilized as a buffer layer to fabricate a flexible, transparent, and stable conductive substrate based on solution-processed AgNWs and graphene under ambient conditions. The transmittance of the AgNWs spin-coated on AGPTi/glass was 99.4% at 550 nm. The high adhesion strength was demonstrated by the 3M tape test; no visible changes in the AgNWs were observed. In addition, AGPTi had an important role in the reductions in the sheet resistances of the pressed AgNWs and graphene on AGPTi, from 5 to 1.5 Ω/sq and from 700 to 600 Ω/sq, respectively. This was mainly attributed to the enhanced connection of wire-to-wire junctions of the AgNWs, and enlargement in the contact area between graphene particles during the pressing process. Furthermore, AGPTi acted as a highly functional buffer layer, absorbing the applied pressure between the conductive materials and rigid substrate, where the AgNWs and graphene particles provided stable structural networks. In order to investigate the applicability of the AGPTi layer in printed electronics, gravure printings of the optimized AgNWs and graphene ink were performed on the flexible substrates at 3 cm/s, yielding line widths of 490 ± 15 µm (gravure-printed AgNWs/AGPTi/PET) and 470 ± 12 µm (gravure-printed graphene/AGPTi/PET), for 500 µm gravure patterns. The good mechanical properties of AGPTi were demonstrated by a bending test (1000 cycles), which showed the high flexibility of the structure with the AGPTi O–I nanohybrid. Therefore, AGPTi as a buffer layer for printed electronics can increase the application of conductive materials to high-end electronic devices requiring high transparencies, flexibilities, and adhesion properties.

## Figures and Tables

**Figure 1 polymers-11-00158-f001:**
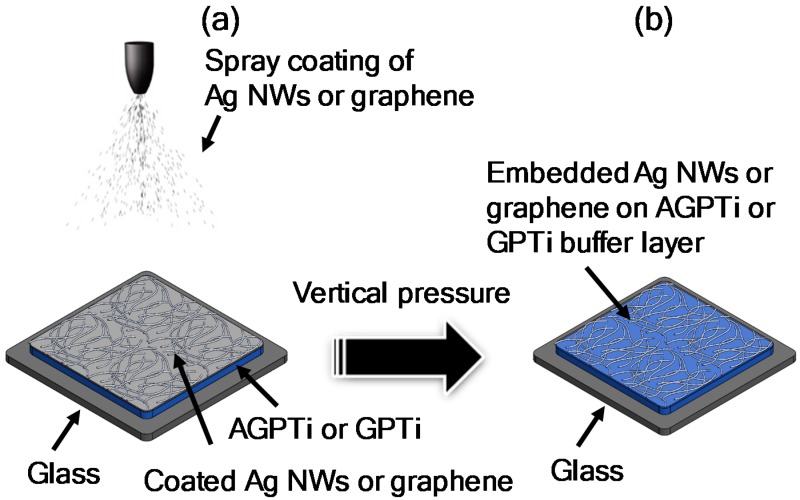
Fabrication of AgNW (or graphene) electrodes with high conductivities and adhesion strengths by mechanical pressing techniques on the AGPTi (or GPTi) buffer layer. (**a**) spray coating of AgNWs or graphene, (**b**) applying pressure to the film.

**Figure 2 polymers-11-00158-f002:**
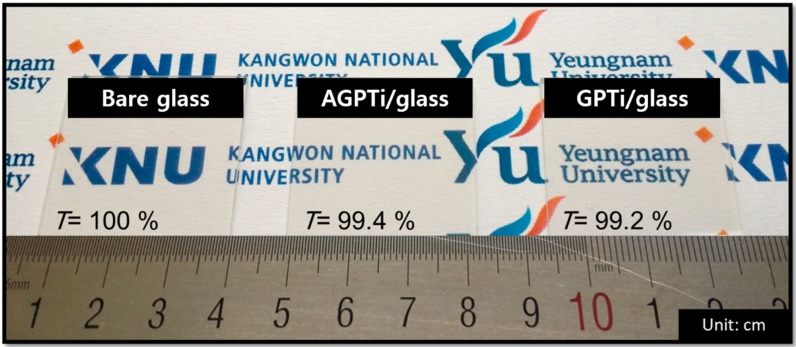
Images of the bare glass and AGPTi- and GPTi-coated glasses used for the fabrication of highly transparent electrodes.

**Figure 3 polymers-11-00158-f003:**
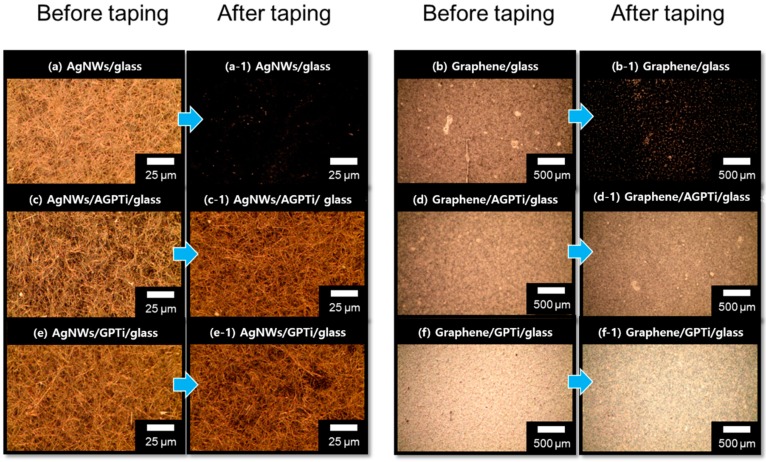
Optical microscopic images before and after the 3M taping tests on AgNWs/bare glass (**a**,**a-1**), graphene/bare glass (**b**,**b-1**), AgNWs/AGPTi (**c**,**c-1**), graphene/AGPTi (**d**,**d-1**), AgNWs/GPTi (**e**,**e-1**), and graphene/GPTi (**f**,**f-1**).

**Figure 4 polymers-11-00158-f004:**
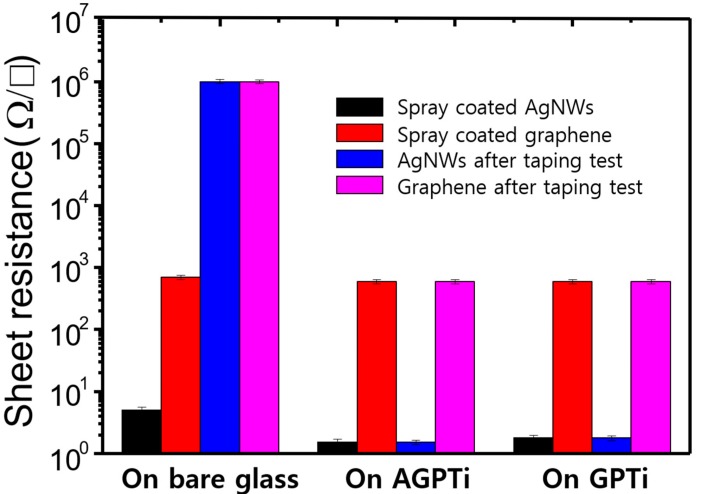
Changes in the sheet resistances of the AgNWs and graphene after the taping tests on the bare glass and AGPTi- and GPTi-coated glasses.

**Figure 5 polymers-11-00158-f005:**
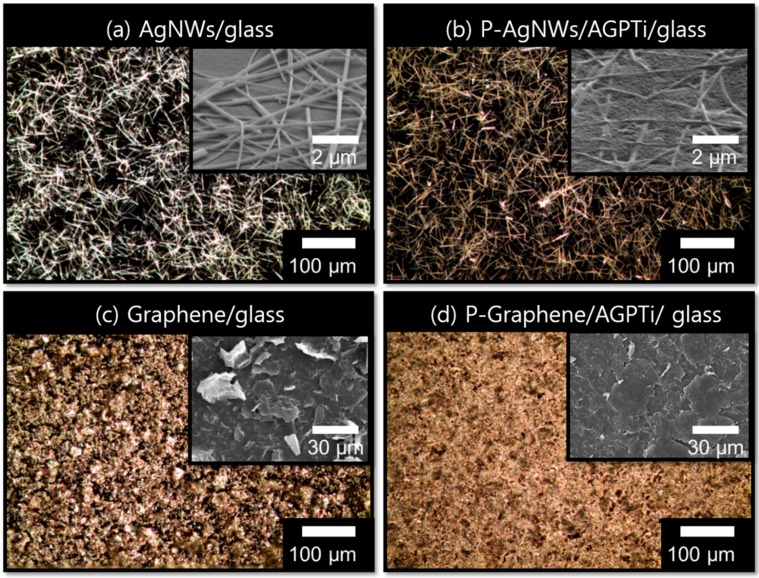
OM and SEM images of the AgNWs and graphene before (**a**,**b**) and after (**c**,**d**) the mechanical pressing, respectively.

**Figure 6 polymers-11-00158-f006:**
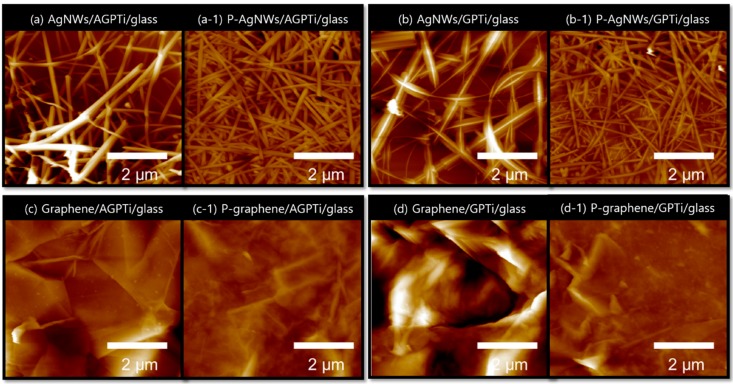
AFM images of the AgNWs and graphene on AGPTi and GPTi before ((**a**) AgNWs/AGPTi/glass, (**b**) AgNWs/GPTi/glass, (**c**) graphene/AGPTi/glass, and (**d**) graphene/GPTi/glass), and after ((**a-1**) P-AgNWs/AGPTi/glass, (**b-1**) P-AgNWs/GPTi/glass, (**c-1**) P-graphene/AGPTi/glass, and (**d-1**) P-graphene/GPTi/glass) the mechanical pressing process.

**Figure 7 polymers-11-00158-f007:**
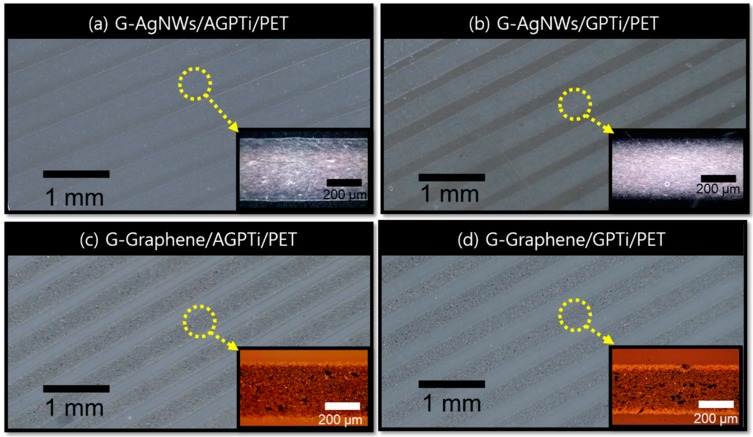
Gravure-printed lines of AgNWs and graphene on AGPTi ((**a**) G-AgNWs/AGPTi/PET and (**c**) G-graphene/AGPTi/PET) and GPTi ((**b**) G-AgNWs/GPTi/PET and (**d**) G-graphene/GPTi/PET), which were predeposited on PET.

**Figure 8 polymers-11-00158-f008:**
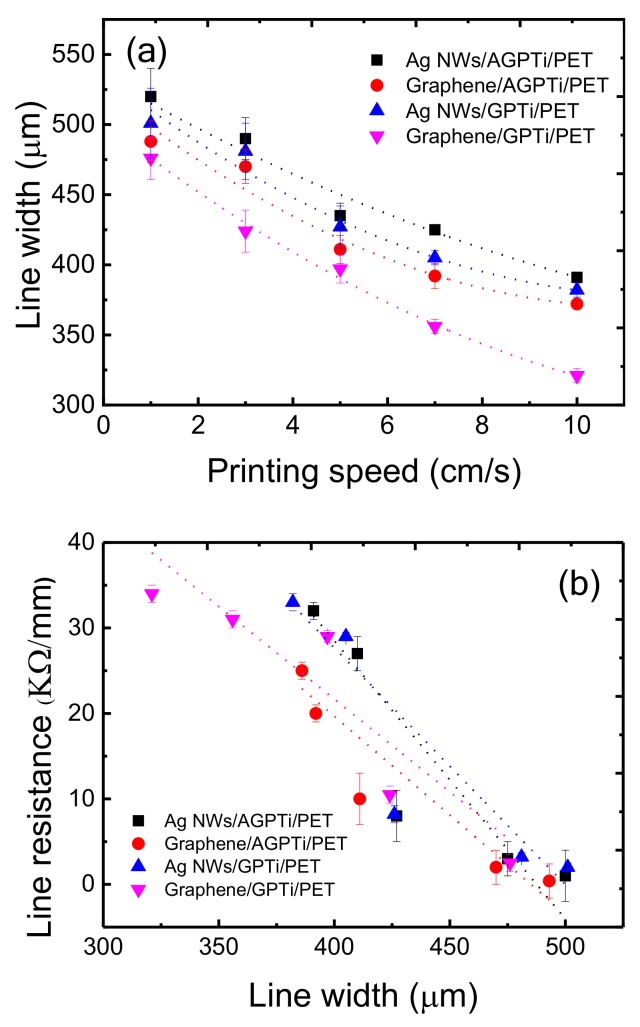
(**a**) Relationship between the printing speed and line width. (**b**) Change in line resistance as a function of the line width.

**Figure 9 polymers-11-00158-f009:**
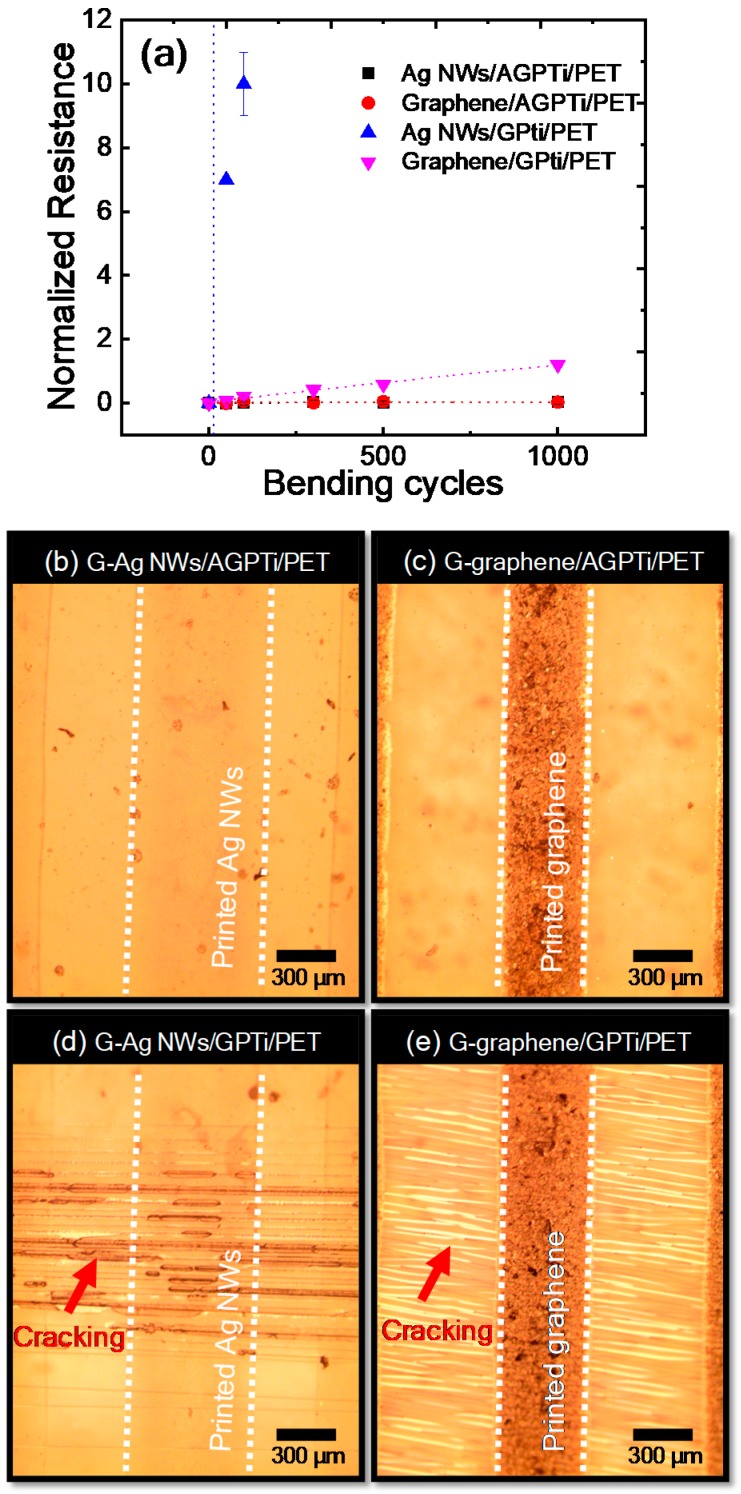
(**a**) Normalized line resistances of the gravure-printed AgNWs and graphene on AGPTi and GPTi as a function of the bending cycle. Optical images of the gravure-printed AgNWs and graphene on AGPTi ((**b**) G-AgNWs/AGPTi/PET and (**c**) G-graphene/AGPTi/PET) and GPTi ((**d**) G-AgNWs/GPTi/PET and (**e**) G-graphene/GPTi/PET) after 1000 bending cycles.

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
