# Peer review of "Sol–Gel-Processed Organic–Inorganic Hybrid for Flexible Conductive Substrates Based on Gravure-Printed Silver Nanowires and Graphene"

_polymers, 2019, doi:10.3390/polym11010158_

Round 1
Reviewer 1 Report
The manuscript reveals synthesis and characterisation of organic-inorganic sol-gel processed flexible conductive substrates. The work is interesting and well realised, but its presentation needs some revision. Comments are given below.
Keywords: “printing” is too general keyword, please add a second word to specify the kind of printing.
SiO2-TiO2 network is not novel since number of manuscripts are on this subject. Please remove word “novel” from the text talking about this network.
Please describe accurately the methods used for characterisation including all adjusted parameters.
Please move the first two paragraphs of Section 3 to Introduction because it does not discuss any results just literature overview.
Fig. 3: Please add to the legend what kind of images are in this Figure, i.e. method used to take images.
Fig. 4: Please leave an empty space in the diagram to be sure that the bars are not hither than given units.
Unit on y axis has some problems, please, check it.
This Figure is not cited in the text.
What kind of tips and mode were used to take AFM images (see comment No. 3).
Author Response
I attached the file for response to review.

Reviewer 2 Report
In this paper, the authors synthesized AGPTi through sol-gel-processed and subsequent steps. As a buffer layer for printed electronics, which can increase the application of conductive materials to high-end electronic devices requiring high transparencies, flexibilities, and adhesion properties. In my view, this present work is interesting. So I would like to recommend the publication of this manuscript after some revision.
The detailed comments can be found below:
1.A space is essential between the numeral and the unit, i.e. 5V should be changed to 5 V. Similar corrections should be done throughout the whole manuscript, including all figures and tables. The only two exceptions are % and oC.
2. The authors think that the synthesis temperature is relatively low, but the reviewer thinks that the authors need to make a comparison with the related published papers in temperature.
3. The description about Fig. 2. in the text is not consistent with the content presented in the figure, please explain or amend.
4. Some grammatical or format errors should be corrected.
Author Response

(The authors gave the same response as above.)
